# Occurrence of *Mycoplasma parvum* in German Pigs of Different Age Groups Using a Novel Quantitative Real-Time PCR Assay

**DOI:** 10.3390/pathogens11111374

**Published:** 2022-11-18

**Authors:** Julia Ade, Katharina Hoelzle, Julia Stadler, Mathias Ritzmann, Ludwig E. Hoelzle

**Affiliations:** 1Institute of Animal Science, Institute of Animal Science, Department of Livestock Infectiology and Environmental Hygiene, University Hohenheim, 70599 Stuttgart, Germany; 2Clinic for Swine, Centre for Clinical Veterinary Medicine, LMU Munich, 85764 Oberschleissheim, Germany

**Keywords:** hemotrophic mycoplasmas, *Mycoplasma parvum*, quantitative PCR, phylogenetics

## Abstract

*Mycoplasma (M.) parvum* is a hemotrophic bacterium circulating in the blood of pigs but is not considered a primary pathogen. Only a handful of studies dealing with this agent have been published since its first description in 1951, and many issues, including epidemiology and the impact of subclinical infections, are yet to be elucidated. This study aimed to establish a *M. parvum* specific real-time PCR for its detection and quantification in porcine blood and the application of this assay to obtain insights into the occurrence of *M. parvum* in German pigs. Furthermore, 16S rDNA amplicons of *M. parvum* positive blood samples were phylogenetically analyzed using MEGA 11 software. The established qPCR targeting the *M. parvum* glyceraldehyde-3-phosphate dehydrogenase encoding gene (*gap*) showed a lower detection limit of 10 gene copies per reaction and no cross-reactivity within the specificity test. A total of 36.0% (*n* = 72) of the sampled fattening pigs, 25.0% (*n* = 15) of the sows, and 4.37% (*n* = 8) of the boars tested *M. parvum* positive. The dendrogram showed the typical allocation of the *M. parvum* isolates into the “haemominutum group” subgroup within the hemotrophic *Mycoplasma* species. Both the novel established qPCR and the obtained epidemiological data can serve as an important basis for future studies dealing with *M. parvum.*

## 1. Introduction

Hemotrophic mycoplasmas (HMs), small cell-wall-less pathogens, are found on and in erythrocytes in the blood of a wide range of mammalian hosts, including human beings [1,2,3]. Three porcine HM species are known so far: the two long-known species *Mycoplasma suis* and *Mycoplasma parvum* [4] and the recently described species *Candidatus* (*Ca*.) Mycoplasma haemosuis [5]. In addition, further porcine HM species or subspecies are suspected but not yet confirmed [6]. In pigs, HM-induced diseases are complex and comprise a variety of clinical signs. *Mycoplasma suis* and *Ca*. M. haemosuis can cause acute diseases that are characterized by anemia, icterus, and high fever. All three HM species are known to induce chronic and persistent infections with low-grade anemia or subclinical courses, effects on growth rates as well as on fattening and reproduction performance [5,7,8,9,10,11,12,13]. Of all three porcine HM species, *M. parvum* is considered the least pathogenic species. Only a few studies concerning *M. parvum* have been published so far [14,15], including the experimental infection of one pig to provide bacteria for sequencing purposes [16]. However, the long-term effects of chronic and subclinical infections with *M. parvum* have not been studied so far. It therefore remains undetermined whether immune-mediating effects are induced in the host during *M. parvum* infection, as is known for *M. suis* [17]. Very little is also known about the prevalence of *M. parvum*. The reason for this is that there are no *M. parvum* specific PCR assays available so far. Evidence about the occurrence of *M. parvum* is obtained from studies investigating all porcine HM species using cross-species PCR assays [5,6,18]. As HMs cannot yet be cultivated in vitro, in the last two decades, specific molecular detection methods for HMs have proven to be the basic prerequisite to obtain insights into the epidemiology, species, and strain differentiation, as well as into the clinical impact of HM infections [9,19,20,21,22,23]. The present study first aimed to develop a specific real-time PCR assay for the quantitative diagnostic of *M. parvum* in pigs. In a second step, this novel qPCR assay was applied to blood samples of domestic pigs from different herds and age groups, i.e., fattening pigs, sows, and boars, to investigate the circulation of *M. parvum* in Germany for the first time ever since the first description of *M. parvum* in 1950 [4]. Furthermore, the genetic diversity of *M. parvum* was compared among seven appropriate isolates by 16S rDNA phylogeny.

## 2. Materials and Methods

### 2.1. Blood Samples and DNA Preparation 

EDTA-anticoagulated blood samples were collected from fattening pigs, sows, and boars from southern Germany (federal states of Baden-Wuerttemberg and Bavaria). Figure 1 illustrates the sampled area of Germany shaded in grey.

Blood samples of the sows (*n* = 60) were collected by the punctuation of the jugular vein. Sows (gilts as well as different parity) originated from a total of three piglet-producing farms with 20 sows per farm sampled. For the group of fattening pigs and boars, DNA extracts from EDTA-anticoagulated blood were available from previous studies [9,24]. Briefly, a total of 200 fattening pigs from 20 farms (10 animals per farm) were sampled at the time of slaughter [9], and 183 boars, housed for artificial insemination in a boar stud and originating from a total of 26 multiplier farms [24], were sampled by punctuation of the jugular vein. The boars were aged between 9 and 77 months. The number of sampled boars per multiplier farm ranged from 1 to 42. None of the animals showed any clinical signs. The fattening and multiplier farms had an unknown HM history and were chosen randomly. As sows were sampled during regular on-farm health monitoring, no ethical approval was needed according to the German Animal Welfare Law. Similarly, sampling of the fattening pigs did not require ethical approval according to the German Animal Welfare Law and the Directive 2010/63/EU, as blood collection was performed after slaughtering [9]. The sampling of the boars was approved by the ethical commission of the veterinary faculty of the Ludwig-Maximilians-Universität, Munich (authorization reference number: 245-17-12-2020) [24]. Bacterial DNA was extracted from the EDTA-anticoagulated blood, as described elsewhere [9,25]. Blood as well as DNA samples were stored at −20 °C prior to use. DNA extracts available from previous studies were submitted for a quality analysis using a NanoDrop^TM^ 2000 (Thermo Fisher Scientific, Dreieich, Germany) before use. 

### 2.2. 16S rDNA and Amplicon Sequencing 

In order to obtain *M. parvum* positive DNA for the establishment of the *M. parvum* specific quantitative real-time PCR as well as for phylogenetical analysis, a total of 100 of the DNA samples were randomly chosen for conventional, HM-specific 16S rDNA using the primers described by Volokhov and coworkers [26]. The 25 µL PCR reaction mixture contained 12.5 µL of Maxima Hot Start Taq DNA-Polymerase (Thermo Fisher Scientific^TM^), 10.5 µL of primer–water mixture (containing 0.5 µmol of the primer UNI_16S_mycF and 0.5 µmol of the primer UNI_16S_mycF [26]), and 2.0 µL of template DNA. Cycling conditions consisted of 95 °C for 3 min and 30 cycles at 95 °C for 15 s and 57 °C for 1 min. The PCR was directly analyzed after the run by means of agarose gel electrophoresis. *Mycoplasma parvum* positive DNA samples were then analyzed with *M. suis* and *Ca*. M. haemosuis specific real-time PCR (see Section 2.4) to exclude *M. suis* and sequences within the 16S rDNA amplicons. Afterward, seven 16S rDNA PCR amplicons from *M. suis* and *Ca*. M. haemosuis negative animals were selected for Sanger sequencing (Seqlab Sequence Laboratories, Göttingen, Germany). Obtained sequences were compared to GenBank entries using the NCBI BLAST tool [27].

### 2.3. Phylogenetic Analysis 

The 16S rDNA amplicons of *M. parvum* (*n* = 7) as well as 16S rDNA amplicons (*M. parvum* and other HM species) were used for a phylogenetic analysis. Table 1 gives an overview of all sequences included in the study.

The 16S rDNA sequences were aligned with the multiple sequence alignment tool of ClustalW in MEGA X [28], and phylogenetic analysis was performed using the neighbor-joining method with a 1000 bootstrap value in MEGA 11 Software [28,29]. 

### 2.4. M. suis and Ca. M. haemosuis qPCR

The *M. suis* status was defined by a specific qPCR targeting the MSG1 gene of *M. suis*, as described elsewhere [25,30]. Likewise, the recently described *Ca.* M. haemosuis qPCR targeting the *Ca. M. haemosuis gap* gene was used for the determination of the *Ca.* M. haemosuis status. While both the *M. suis* and *Ca.* M. haemosuis status of the fattening pigs and boars was ascertained in previous studies [9,24], qPCR of the sows was performed in the present study. 

### 2.5. Primer Design

The following primers targeting the *gap* gene of *M. parvum* were designed with Primer 3 software [31,32]. *MPa*F`- 5′-ATGCTGGCGCTCCTAAAGTT-3′ and *MPa*R´- 5′-CTGCTGCAGCTCTAGCTCTT-3′. *gap* encodes the glyceraldehyde-3-phosphate dehydrogenase (GAPDH) of *M. parvum* (accession no. NC_022575.1). Test specificity was analyzed in silico by using an NCBI-provided BLAST algorithm as well as by sequencing (Seqlab Sequence Laboratories, Göttingen, Germany) of the resulting 276 bp *gap* PCR fragment of *M. parvum*. Sequences were compared to GenBank entries using the BLAST tool provided by NCBI.

### 2.6. Cloning of the qPCR Fragment (qPCR-Positive Control) 

*Escherichia coli* transformants (pC_Parvum) containing the 276 bp qPCR *gap* fragment of *M. parvum* were produced by cloning the amplicon into the plasmid vector pCR2.1 (Invitrogen) according to the manufacturer’s instructions. Plasmid DNA was extracted and purified from the *E. coli* transformants (pC_Parvum) with the GenElute^TM^ Plasmid Miniprep Kit (Sigma-Aldrich, Taufkirchen, Germany) and obtained plasmid DNA was quantified afterward by means of spectrophotometry (NanoDrop^TM^ 2000, Thermo Fisher Scientific). With a plasmid size of 4.106 kb concentrations, genome equivalents (GE) of *M. parvum* were calculated, and DNA concentrations were adjusted to 45 pg/2 µL representing 1 × 10^7^ GE (http://cels.uri.edu/gsc/cndna.html accessed on 13 November 2022). 

### 2.7. Quantitative Real-Time PCR 

The 20 µL reaction mixture containing 10 µL of the 2x SYBR^®^ Green PCR Master Mix (Thermo Fisher Scientific), 8 µL of primer mixture (containing 0.5 µM primer each), and 2 µL of template DNA was applied to the StepOne^™^ System (Applied Biosystems) for the detection of *M. parvum* DNA. Cycling conditions consisted of 95 °C for 5 min, 40 cycles at 95 °C for 15 s, and 60 °C for 1 min, followed by a melting curve analysis. Quantification was performed by the inclusion of plasmid pC_Parvum DNA standard dilutions (450 fg/2 µL, 45 fg/2 µL, and 4.5 fg/2 µL) representing 10^5^, 10^4^, and 10^3^ GE per reaction run, respectively. PCR results were analyzed using StepOne™ Software Version 2.2 (Applied Biosystems, Dreieich, Germany) and Microsoft^®^ Excel, 2016.

For the determination of the analytical specificity of the novel qPCR assay, DNA from the porcine HM species *M. suis*, *Ca*. M. haemosuis, other hemotrophic and nonhemotrophic *Mycoplasma* spp. (i.e., *M. wenyonii*, *Ca*. M. haemobos, *M. haemofelis*, *M. bovis*, and *M. bovoculi*), and the following panel of other porcine pathogens was used as template DNA: *Escherichia coli* (*n* = 2), *Salmonella* Typhimurium (*n* = 2), *Staphylococcus aureus*, *Pasteurella multocida,* and *Streptococcus suis*. DNA isolation and/or cultivation of the bacteria was performed as previously described [9,25,30,33]. DNA was adjusted to 10^5^ bacteria/ reaction.

Ten-fold dilutions from 10^7^ to 1 GE/reaction out of the pC_Parvum plasmid DNA were analyzed by qPCR to determine the lower limit of detection (LOD).

### 2.8. Intra- and Interassay Repeatability 

The intra-assay repeatability was determined by analyzing four replicates of the plasmid dilutions (10^7^ to 1 GE/reaction) in the same run. The interassay variability was determined by running quadruples of the same plasmid dilution in five different runs on four consecutive days.

## 3. Results

### 3.1. Identification of Mycoplasma parvum Positive Samples

Porcine blood samples (*n* = 100) were analyzed by an HM-specific conventional 16S rDNA PCR, and *M. suis* and *Ca.* M. haemosuis specific real-time PCR. Seven randomly chosen HM-specific 16S rDNA amplicons derived from *M. suis* and *Ca.* M. haemosuis negative pigs were analyzed after Sanger sequencing with the BLAST algorithm. All amplicons showed 99.8–100.0% identity to other GenBank entries for *M. parvum* 16S rDNA genes and could therefore be used as positive controls for the novel *M. parvum* specific real-time PCR. Sequences were uploaded to the GenBank (accession numbers ON703815–ON703821).

### 3.2. Development of a M. parvum-Specific Quantitative Real-Time PCR 

We developed a quantitative SYBR^®^ green real-time PCR as a diagnostic tool for the detection of *Mycoplasma parvum* in the blood of infected pigs. For this purpose, primers targeting the GAPDH of *M. parvum* were chosen. The novel real-time qPCR assay successfully amplified each of the seven *M. parvum* DNA samples. The specificity of the qPCR was proven because no cross-reactivity was observed with DNA samples from other HM species and nonhemotrophic mycoplasma species or from other porcine bacterial pathogens. 

Melting curve analyses revealed a *M. parvum* specific melting temperature of 74.177 °C (±0.135 °C). Ten-fold serial dilutions of the plasmid pC_Mparvum from 45 pg to 45 ag (corresponding to 10^7^ to 10^1^ genome equivalents) per reaction were used to determine the analytical sensitivity of the qPCR assay. All different concentrations in the standard dilutions from 10^7^ to 10^1^ GE were consistently detected by the qPCR assay. The highest dilution yielding consistently positive qPCR results contained 0.45 fg pC_Mparvum DNA per reaction corresponding to 10 GE per reaction. The lower limit of detection of the qPCR was thus 10 *gap* copies per reaction and 2.5 × 10^3^ GE per mL blood. 

The intra- and interassay repeatability is shown in Table 2. 

### 3.3. Hemotrophic mycoplasma Infection in Fattening Pigs, Sows, and Boars

Using the herein established, novel established, *M. parvum*-specific qPCR assay, *M. parvum* could be detected in the blood of each sampled age group of fattening pigs, sows, and boars. Regarding the fattening pigs, 72 out of 200 pigs were positive for *M. parvum,* whereas 38 and 35 out of the 200 were positive for *M. suis* and *Ca*. M. haemosuis, respectively. *M. parvum* was present in the blood of 15 out of the 60 sampled sows. *M. suis* and *Ca.* M. haemosuis were detected in 4 and 13 sows, respectively. A total of 8 out of the 183 boars were positive for *M. parvum* in qPCR, whereas the other two HM species could not be detected in any blood sample of this age group. Co-infections with three HM species were detected in fattening pigs, co-infections with two HM species were detected in fattening pigs as well as in sows.

Average bacterial blood-loads equaled 9.34 × 10^6^
*M. parvum*/mL blood, 7.62 × 10^7^
*M. suis*/mL blood and 1.61 × 10^5^
*Ca*. M. haemosuis/mL blood in fattening pigs; 1.18 × 10^6^
*M. parvum*/mL blood, 3.58 × 10^4^
*M. suis*/mL blood and 4.83 × 10^5^
*Ca*. M. haemosuis/mL blood in sows; and 1.12 × 10^5^
*M. parvum*/mL blood in boars. 

Table 3 gives a detailed overview of the HM status of all investigated samples. 

### 3.4. Mycoplasma parvum Phylogeny 

Seven 16S rDNA *M. parvum* amplicons derived from samples in the present study were used for phylogenetic analyses. Those seven amplicons were randomly chosen from seven different herds without any known contact with each other. Furthermore, the 16S rDNA sequences of *M. parvum*, *M. suis*, *Ca*. M. haemosuis, *M. wenyonii*, *M. ovis*, *M. coccoides*, *M. haemofelis*, and *Ca*. M. haemominutum were obtained from GenBank and are included in the dendrogram. The neighbor-joining method with a 1000 bootstrap value was chosen. The dendrogram is shown in Figure 2. 

The obtained *M. parvum* isolates clustered within the “hemominutum” group of the HM species in close relation to *M. suis*. Among the novel German *M. parvum* isolates (accession numbers ON703815–ON703821), two subclusters (orange and yellow highlighted clades) could be distinguished. The *M. parvum* isolate obtained in Brazil (accession number MK287838.1) could be assigned to a further cluster (blue highlighted clade) within all the *M. parvum* isolates.

## 4. Discussion

The study herein described the occurrence of *Mycoplasma parvum* in fattening pigs, sows, and boars in Germany, for which a novel quantitative real-time PCR was successfully established. 

*Mycoplasma parvum* is a long-known but rarely described and investigated porcine HM species. To the best of our knowledge, the novel qPCR assay is the first quantitative and specific assay detecting *M. parvum* in pigs. We used this specific qPCR assay to quantitatively detect *M. parvum* in the blood samples of fattening pigs, sows, and boars from southern Germany. So far, the few descriptions of *M. parvum* reported were based on 16S rDNA PCR or microscopy [5,14,15,16]. Our established qPCR assay specifically targets the *gap* gene of *M. parvum,* which encodes the GAPDH of *M. parvum*. Glyceraldehyde-3-phosphate dehydrogenase encoding genes have successfully been used as target genes in diagnostics of HMs before, such as for the identification of *M. suis*, *Ca*. M. haemosuis, *M. wenyonii,* and *Ca*. M. haemobos [9,30,33]. Often, the 16S rDNA gene is chosen as the PCR target gene because it is considered highly conserved within bacterial species. However, in the case of the porcine HM species, the 16S rRNA gene is not well suited due to the low differentiation potential, as the 16S rDNA genes of *M. suis* and *M. parvum* show nearly 96% identity among each other [16]. In contrast, the GAPDH-encoding genes of *M. suis* and *M. parvum* have only 76% identity. Therefore, we decided to use the gap gene as a PCR target, which is also a housekeeping gene and known to be highly conserved within bacterial species but significantly different between *M. parvum* and *M. suis.* The successful target selection was also reflected in the determination of the analytical specificity. Analytical specificity was first predicted through in silico analysis and verified afterward by negative qPCR results with DNA of other *Mycoplasma species* and further porcine bacterial pathogens. 

In the few previous studies dealing with *M. parvum*, a quantification of bacterial blood loads was not performed. Within the herein described novel *M. parvum* qPCR assay, quantification with a sensitivity of 10 genome equivalents per PCR, corresponding to 2.5 × 10^3^ bacteria per mL blood, is possible. This is in line with similar qPCR assays of other HM species such as *M. suis*, *Ca*. M. haemosuis, *M. wenyonii*, *Ca*. M. haemobos, *M. haemocanis,* and *Ca*. M. haematoparvum [9,30,33,34], and displays a high analytical sensitivity. Therefore, we assume that our novel qPCR assay allows the identification of asymptomatic infected carrier animals. This was evidenced by the application of the qPCR for the successful detection of *M. parvum* in fattening pigs, sows, and boars. To the best of our knowledge, this was the first survey for a European sample panel within different age groups and 443 tested pigs in total. *M. parvum* was present in each of the sampled age groups, i.e., in fattening pigs, sows, and boars. Interestingly, the *M. parvum* infection rate was significantly higher in fattening pigs and sows with 36.0% and 25.0% of *M. parvum* positive animals, respectively, compared with boars, with a total of 4.37% *M. parvum* positive animals.

There could have been many reasons for the variation in the infection rates among the different age groups and could especially include potentially higher biosecurity levels in boar studs, as well as individually housed animals and the absence of group vaccinations in boar studs compared with fattening and piglet-producing farms. Furthermore, the higher HM rates in sows compared with boars could be owing to the known general changes and depression of the immune system during pregnancy [35,36,37]. The occurrence of *M. parvum* in German fattening pigs and sows in the present study was lower compared with that in Brazil, with 79.7% positive sows [38], and lower than in China, with 50.0% *M. parvum* positive pigs (feeder pigs and sows) [5]. On the contrary, studies in Japan and South Korea revealed lower *M. parvum* prevalence, with 15.0% (feeder pigs and sows) and 2.7% positive pigs (various age groups), respectively [7,39]. For boars, however, there are no comparable studies dealing with the prevalence of *M. parvum* in this age group. Comparing the detection rates of the different HM species within the sampled animals, *M. parvum* was more frequent in each of the age groups than *M. suis* and *Ca*. M. haemosuis. A higher prevalence of *M. parvum* compared with *M. suis* in the same animals was also reported by Seo and coworkers in South Korea [7] and by Gatto and coworkers in Brazil [38]. 

Coinfections with two or three HM species are a commonly observed issue and have been previously described in pigs, for example, in China, with coinfections of *M. suis*/*M. parvum* and the novel *Ca. M. haemosuis* [5], as well as in Germany with *M. suis* and *Ca. M. haemosuis* [9]. This phenomenon has further been described in other species such as sheep [40], goats [41], cattle [22,33], dogs [42], and cats [43,44]. 

Despite the long history of *M. parvum*, knowledge of its pathogenicity, and especially its impact on the immune system, is rather limited. So far, two studies conducted 60–70 years ago reported clinical signs similar to *M. suis* induced IAP within experimental *M. parvum* infection [14,15]. From today’s point of view, *M. parvum* seems primarily apathogenic to pigs due to the lack of major clinical field cases and the lack of clinical signs in a current experimental infection study conducted in one splenectomized pig [16]. On the contrary, *M. parvum* was related to the loss of performance in subclinical infected pigs in Brazil [10]. In the present study, *M. parvum* was also frequently detected in clinically healthy animals in various age groups in Germany. Chronical and subclinical infections have extensively been described in pigs with *M. suis* [25,45,46] and *Ca.* M. haemosuis [5,7,9,24]. For *M. suis*, it is known that such infections are related to various silent influences, especially those related to dysregulation of the immune system, a higher susceptibility to other infections, and impacts on reproductive performance (e.g., increased stillbirth rates) [17,25,45,46,47]. Thus, such subclinical *M. suis* infections are of major economic importance to the porcine industry. Whether *M. parvum* is also capable of causing such effects in the course of subclinical infections has not yet been investigated. In the context of the widely observed subclinical infections of *M. parvum* found in the present study, this should be a research topic to be investigated in the future. Our herein developed novel *M. parvum* specific qPCR assay was proven to be a suitable and reliable diagnostic tool to implement in future studies.

## 5. Conclusions

In this study, a quantitative real-time PCR assay was established for the diagnosis of *M. parvum* in blood samples of pigs. Applying this PCR assay to blood samples of different-aged German pigs, *M. parvum* could successfully be detected in 36.0% (*n* = 72) of fattening pigs, in 25.0% (*n* = 15) of sows, and in 4.27% (*n* = 8) of boars. Both data on the occurrence of *M. parvum* in Germany as well as the novel qPCR are a profound basis for the further studies required to evaluate the medical and economic significance of *M. parvum* in pigs. 

## Figures and Tables

**Figure 1 pathogens-11-01374-f001:**
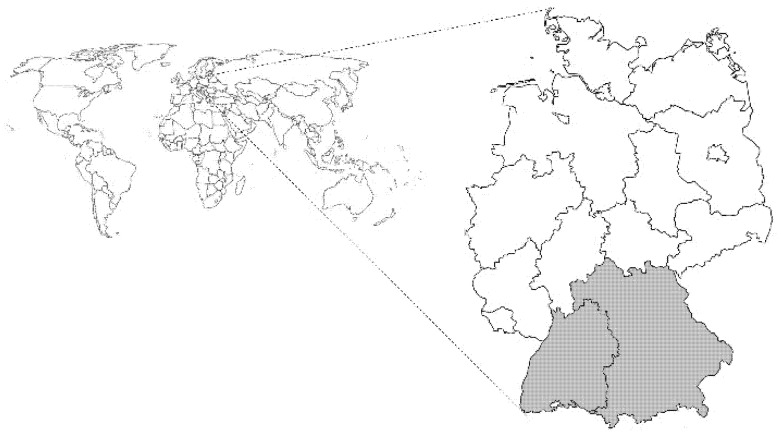
Sampling areas: the grey-shaded areas show the federal states of Baden-Wuerttemberg and Bavaria (Germany), from which the sampled animals originated.

**Figure 2 pathogens-11-01374-f002:**
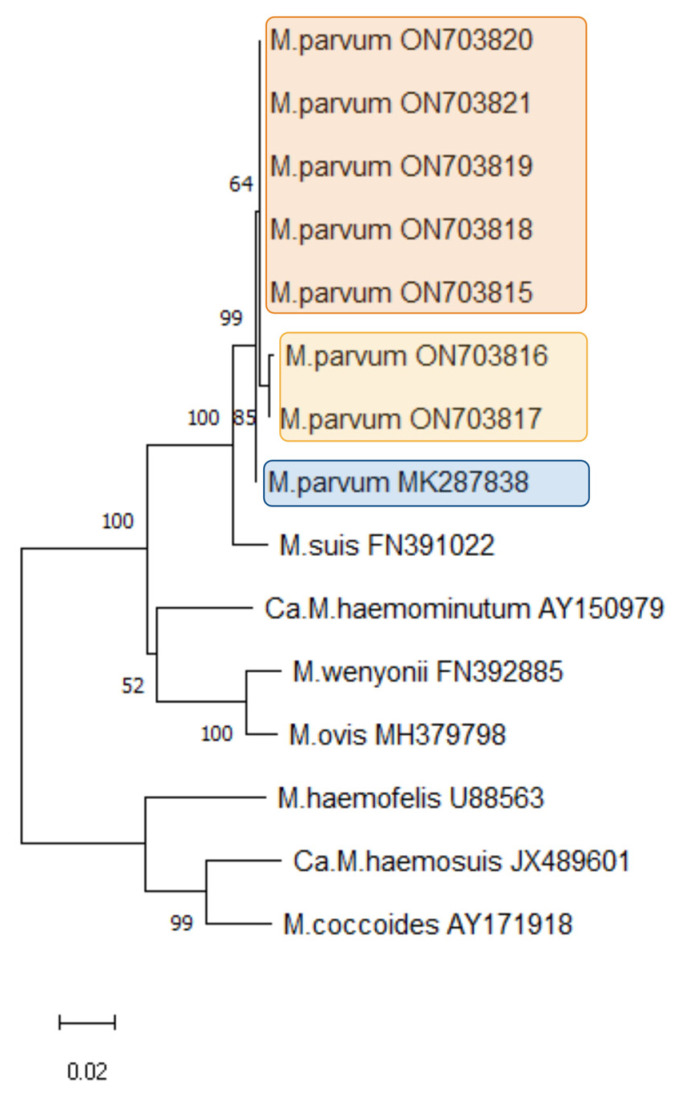
The 16S rDNA phylogenetic analysis of *M. parvum* isolates obtained in the present study (created with MEGA 11 software).

**Table 1 pathogens-11-01374-t001:** Sequences of the different HM species included in the phylogenetic analysis.

HM Species and Name in the Phylogenetic Tree	Acc. Number (Version)	Host Species	Country of Origin
** *Mycoplasma parvum* **			
*M. parvum* ON703815 *	ON703815.1	pig	Germany
*M. parvum* ON703816 *	ON703816.1	pig	Germany
*M. parvum* ON703817 *	ON703817.1	pig	Germany
*M. parvum* ON703818 *	ON703818.1	pig	Germany
*M. parvum* ON703819 *	ON703819.1	pig	Germany
*M. parvum* ON703820 *	ON703820.1	pig	Germany
*M. parvum* ON703821 *	ON703821.1	pig	Germany
*M. parvum* MK287838	MK287838.1	pig	Brazil
*M. suis* FN391022	FN391022.1	pig	Germany
*Ca*. M. haemosuis JX489601	JX489601.1	pig	China
*M. wenyonii* FN392885	FN392885.1	cattle	Germany
*M. ovis* MH379798	MH379798.1	sheep	Brazil
*M. coccoides* AY171918.1	AY171918.1	mouse	USA
*M. haemofelis* U88563	U88563.2	cat	USA
*Ca*. M. haemominutum AY150979	AY150979.1	cat	South Africa

* Sequences acquired in this study (highlighted in grey).

**Table 2 pathogens-11-01374-t002:** Intra- and interassay repeatability values of the novel established qPCR.

	Intra-Assay Repeatability	Interassay Repeatability
Plasmid Copy Number	Mean CT (±SD) *	Mean CT (±SD)
10^7^	15.23 (±0.44)	15.02 (±0.92)
10^6^	19.01 (±0.27)	18.94 (±0.92)
10^5^	22.78 (±0.11)	21.92 (±0.94)
10^4^	26.20 (±0.15)	25.25 (±0.96)
10^3^	29.93 (±0.07)	29.28 (±0.75)
10^2^	32.81 (±0.73)	31.89 (±0.99)
10^1^	35.92 (±0.81)	35.51 (±0.54)

* Mean threshold cycle values and standard deviations.

**Table 3 pathogens-11-01374-t003:** An overview of the quantitative detection of HMs in sampled fattening pigs, sows, and boars.

				Fattening Pigs		
				*M. parvum* ^1^	*M. suis* ^2^	*Ca*. M. haemosuis ^2^
qPCR positive pigs (total)				72/200 (36.00%)	38/200 (19.00%)	35/200 (17.50%)
bacterial blood-loads (per mL blood)			Average:	9.34 × 10^6^	7.62 × 10^7^	1.61 × 10^5^
			Range:	3.41 × 10^3^–4.82 × 10^8^	9.58 × 10^2^–1.14 × 10^9^	5.52 × 10^3^–1.55 × 10^6^
Co-infections						
*M. parvum*	+	*M. suis*	17/200 (8.50%)			
*M. parvum*	+	*Ca*. M. haemosuis	19/200 (9.50%)			
*M. suis*	+	*Ca*. M. haemosuis	12/200 (6.00%)			
all species (triple infection)			5/200 (2.50%)			
				**Sows**		
				*M. parvum* ^1^	*M. suis* ^1^	*Ca*. M. haemosuis ^1^
qPCR positive pigs (total)				15/60 (25.00%)	4/60 (6.67%)	13/60 (21.67%)
bacterial blood-loads (per mL blood)			Average:	1.18 × 10^6^	3.58 × 10^4^	4.83 × 10^5^
			Range:	5.48 × 10^3^–1.53 × 10^7^	5.66 × 10^3^–5.49 × 10^4^	2.39 × 10^3^–3.56 × 10^6^
Co-infections						
*M. parvum*	+	*M. suis*	0/60 (0.00%)			
*M. parvum*	+	*Ca*. M. haemosuis	3/60 (5.00%)			
*M. suis*	+	*Ca*. M. haemosuis	1/60 (1.66%)			
all species (triple infection)			0/60 (0.00%)			
				**Boars**		
				*M. parvum* ^1^	*M. suis* ^3^	*Ca*. M. haemosuis ^3^
qPCR positive pigs (total)				8/183 (4.37%)	0/183 (0.00%)	0/183 (0.00%)
bacterial blood-loads (per mL blood)			Average:	1.12 × 10^5^	/	/
			Range:	6.23 × 10^3^–6.26 × 10^6^	/	/
Co-infections						
*M. parvum*	+	*M. suis*	0/183 (0.00%)			
*M. parvum*	+	*Ca*. M. haemosuis	0/183 (0.00%)			
*M. suis*	+	*Ca*. M. haemosuis	0/183 (0.00%)			
all species (triple infection)			0/183 (0.00%)			

^1^ determined in the present study; ^2^ determined in the study of Ade and co-workers (2022) [9]; ^3^ determined in the study of Ade and co-workers (2021) [24].

## Data Availability

Not applicable.

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
