# Peer review of "Occurrence of Mycoplasma parvum in German Pigs of Different Age Groups Using a Novel Quantitative Real-Time PCR Assay"

_pathogens, 2022, doi:10.3390/pathogens11111374_

Round 1
Reviewer 1 Report
Dear authors,
Interesting paper. I have some comments for you to discuss.
Epidemiological studies on Mycoplasma parvum in fattening 2 pigs, sows and boars in Germany.
The title does not cover/reflect the content of the paper. what you present in the paper is described in line 13-15. With the exception that samples are not taken randomly in Germany. So better adjust title to south Germany and something about the PCR.
Line 44: Evidence of what exists?
Line 67: What is the background of these boars. How many herds? (Adult) reproduction boars used for fertilization of sows, natural breeding, AI centers? Or are this boars not used for fertilization, only for heat detection?
Line 70 how many boars were sampled per multiplier herd?
line 195: in the text it says 9 boars and in the table it says 8 boars
Line 208 were these samples chosen from different originating herds? or were these samples via animal transport connected?
Line 232-234: Is it possible that due to the sampling group these sequences show a high agreement. Could this be different in populations in other countries or maybe the America’s an/or Asian countries and what implications will this have on your primer selection?
Biosecurity is not the only feature different between sow and fattening herds and boar studs. Animals are housed individually and probably also less group vaccinations.
Line 253: in here it becomes clear where the samples of the boars originate from. Please ad this in M&M. Can you also share something about the age of the sows and boars?
Author Response
The title does not cover/reflect the content of the paper. what you present in the paper is described in line 13-15. With the exception that samples are not taken randomly in Germany. So better adjust title to south Germany and something about the PCR.
We agree with the reviewers comment and changed the title to “Occurrence of Mycoplasma parvum in German pigs of different age groups using a novel quantitative real-time PCR assay”.
Line 44: Evidence of what exists?
Thank you for this comment. We intended to express that evidences for the occurrence of M. parvum are existing. We adapted the sentence accordingly.
Line 67: What is the background of these boars. How many herds? (Adult) reproduction boars used for fertilization of sows, natural breeding, AI centers? Or are this boars not used for fertilization, only for heat detection?
Boars are housed for artificial insemination (boar studs). We added this to the manuscript.
Line 70 how many boars were sampled per multiplier herd?
Between 1-42 boars were sampled per multiplier herd. All boars were stalled at a boar stud. We added the number of boars per herd to the manuscript.
line 195: in the text it says 9 boars and in the table it says 8 boars
Thank you for the detection of this typo. We corrected the number to 8.
Line 208 were these samples chosen from different originating herds? or were these samples via animal transport connected?
The samples were chosen from different herds without any known contact between each other. We added this to the manuscript.
Line 232-234: Is it possible that due to the sampling group these sequences show a high agreement. Could this be different in populations in other countries or maybe the America’s an/or Asian countries and what implications will this have on your primer selection?
The phylogeny analysis is based on 16S rDNA sequences, the novel q PCR assay on the GAPDH encoding gene gap. At present, it is not possible to assess whether GAPDH encoding genes differ greatly between isolates of different regional origins (there are no sequences in the database, the sequences deposited in NCBI for M. parvum gap are clearly M. suis gap sequences).
However, for M. suis it is known that the gap sequences from Germany, USA and China have at least 98% identity. Therefore, the probability that this is the case for M. parvum is also relatively high.
Biosecurity is not the only feature different between sow and fattening herds and boar studs. Animals are housed individually and probably also less group vaccinations.
We agree to this valid note and added this topic into our discussion.
Line 253: in here it becomes clear where the samples of the boars originate from. Please ad this in M&M. Can you also share something about the age of the sows and boars?
We added information on the origin of the boars to M&M as described above. We also added information about the age of the sows and boars to the M&M part of the manuscript.
Reviewer 2 Report
The present study deals with an interesting topic and is of general interest to the readers of the journal: an epidemiological study of Mycoplasma parvum, which was first described in 195 and for which there have been few studies. In addition, the authors have developed a specific real-time PCR assay for the quantitative diagnosis of M. parvum in pigs and for the first time studied the circulation of M. parvum in Germany.
I am convinced that the authors have put a lot of effort into this study. The manuscript is very well written and structured. I congratulate the authors and have only minor suggestions.
Yours sincerely
Summary: include n with the %
Check line 77 to see if LMU is declared.
Line 98: reference is needed for the NCBI Blast Tool
Table 2: I was wondering if ANOVA could be amplifying the differences in means here.
Line 190: I suggest avoiding the use of "our" or "we". Use "the study."
In the discussion, I suggest writing the first paragraph with a summary of the main findings of the study.
The same applies to the conclusion section.
Delete (or modify) the Acknowledgements section.
Author Response
Summary: include n with the %
Thank you for this comment. We integrated those data into the abstract.
Check line 77 to see if LMU is declared.
We changed this into “Ludwig-Maximilians-Universität”.
Line 98: reference is needed for the NCBI Blast Tool.
We added a reference (Altschul et al. 1990) to cite the NCBI Blast Tool.
Table 2: I was wondering if ANOVA could be amplifying the differences in means here.
Thank you for this valid comment and approach. To the best of our knowledge, ANOVA analysis compares mean values between three or more groups. However, in our case it would make no sense to compare the mean CT values between different plasmid dilutions.
Line 190: I suggest avoiding the use of "our" or "we". Use "the study."
Thank you for this comment. We amended the sentence accordingly.
In the discussion, I suggest writing the first paragraph with a summary of the main findings of the
study. The same applies to the conclusion section.
We amended the discussion as well as the conclusion accordingly.
Delete (or modify) the Acknowledgements section.
We modified the section accordingly.
Reviewer 3 Report
The importance of this manuscript lies in the absence of molecular tests for the detection of pathogens that generally affect pigs. The authors present interesting data on epidemiology in a specific region of Germany, and through a PCR test, they detect pathogens such as M. parvum, among others. The study is well designed, and the specificity and repeatability tests are well performed.
On the other hand, some points should be addressed by the authors:
- The authors should explain the selection of the use of the gap gene as a molecular marker in their qPCR test. Although they mention the identity percentage of 16% for the 16S rDNA, they need to delve into the decision on the use of gap, which is also a very conserved gene among species. Not to mention that many more molecular markers could also be used.
- The authors have wondered if susceptibility to infection or co-infection by M. parvum, M. suis, Ca. M. haemosuis can be associated with a hormonal issue in sows? This is an interesting point to discuss.
- The quality of the phylogenetic tree could be improved. The authors highlighted the presence of the possible clades for M. parvum; they could be presented in different colors to reinforce the idea.
Author Response
The authors should explain the selection of the use of the gap gene as a molecular marker in their qPCR test. Although they mention the identity percentage of 16% for the 16S rDNA, they need to delve into the decision on the use of gap, which is also a very conserved gene among species. Not to mention that many more molecular markers could also be used.
Thank you for the comment. We rewrote the discussion part and added some information regarding the selection of gap as PCR target.
(“Glyceraldehyde-3-phosphate dehydrogenase encoding genes have successfully been used as target genes in diagnostics of HMs before, such as for the identification of M. suis, `Ca. M. haemosuis`, M. wenyonii and `Ca. M. haemobos` [9,30,33]. Often, the 16S rDNA gene is chosen as PCR target gene because it is considered highly conserved within bacterial spe-cies. However, in the case of the porcine HM species the 16S rRNA gene is not well suited due to the low differentiation potential as the 16SrDNA genes of M. suis and M. parvum show nearly 96% identity among each other [16]. In contrast, the GAPDH-encoding genes of M. suis and M. parvum have only 76% identity. Therefore, we decided to use the gap gene as a PCR target, which is also a housekeeping gene and known to be highly conserved within bacterial species but significantly different between M. parvum and M. suis.”)
The authors have wondered if susceptibility to infection or co-infection by M. parvum, M. suis, Ca. M. haemosuis can be associated with a hormonal issue in sows? This is an interesting point to discuss.
Thank you for this comment. Indeed, this is an interesting and valid point to consider. We added this to our discussion.
(“Reasons for the variation in the infection rates among the different age groups can be miscellaneous and may especially include potential higher biosecurity levels in boar studs as well as individually housed animals and the absence of group vaccinations in boar studs compared to fattening and piglet producing farms. Further, higher HM rates in sows compared to boars could be owed to the known general changes and depression of the immune system during pregnancy [35-37].”)
The quality of the phylogenetic tree could be improved. The authors highlighted the presence of the possible clades for M. parvum; they could be presented in different colors to reinforce the idea.
We agree. We amended the figure to highlight the observed clusters withing M. parvum and mentioned this in the text.